# Thymoquinone: Review of Its Potential in the Treatment of Neurological Diseases

**DOI:** 10.3390/ph15040408

**Published:** 2022-03-27

**Authors:** Faheem Hyder Pottoo, Abdallah Mohammad Ibrahim, Ali Alammar, Rida Alsinan, Mahdi Aleid, Ali Alshehhi, Muruj Alshehri, Supriya Mishra, Noora Alhajri

**Affiliations:** 1Department of Pharmacology, College of Clinical Pharmacy, Imam Abdul Rahman Bin Faisal University, Dammam 31441, Saudi Arabia; 2170005364@iau.edu.sa (A.A.); 2170006017@iau.edu.sa (R.A.); 2170000709@iau.edu.sa (M.A.); maaalshehri@iau.edu.sa (M.A.); 2Department of Fundamentals of Nursing, College of Nursing, Imam Abdul Rahman Bin Faisal University, Dammam 31441, Saudi Arabia; 3College of Medicine and Health Sciences, Khalifa University, Abu Dhabi P.O. Box 127788, United Arab Emirates; 100042658@ku.ac.ae; 4SRM Modinagar College of Pharmacy, SRM Institute of Science and Technology, Delhi-NCR Campus, Modinagar, Ghaziabad 201204, UP, India; supriya.initative@gmail.com; 5Department of Medicine, Sheikh Shakhbout Medical City (SSMC), Abu Dhabi P.O. Box 127788, United Arab Emirates; nalhajri007@gmail.com

**Keywords:** thymoquinone, anti-inflammation, Alzheimer’s disease, neuroinflammation, Parkinson’s disease, epilepsy, mitochondrial dysfunction

## Abstract

Thymoquinone (TQ) possesses anticonvulsant, antianxiety, antidepressant, and antipsychotic properties. It could be utilized to treat drug misuse or dependence, and those with memory and cognitive impairment. TQ protects brain cells from oxidative stress, which is especially pronounced in memory-related regions. TQ exhibits antineurotoxin characteristics, implying its role in preventing neurodegenerative disorders such as Alzheimer’s disease and Parkinson’s disease. TQ’s antioxidant and anti-inflammatory properties protect brain cells from damage and inflammation. Glutamate can trigger cell death by causing mitochondrial malfunction and the formation of reactive oxygen species (ROS). Reduction in ROS production can explain TQ effects in neuroinflammation. TQ can help prevent glutamate-induced apoptosis by suppressing mitochondrial malfunction. Several studies have demonstrated TQ’s role in inhibiting Toll-like receptors (TLRs) and some inflammatory mediators, leading to reduced inflammation and neurotoxicity. Several studies did not show any signs of dopaminergic neuron loss after TQ treatment in various animals. TQ has been shown in clinical studies to block acetylcholinesterase (AChE) activity, which increases acetylcholine (ACh). As a result, fresh memories are programmed to preserve the effects. Treatment with TQ has been linked to better outcomes and decreased side effects than other drugs.

## 1. Introduction

Nigella sativa (fennel flower, black seed) had been used as natural medicine traditionally in the Middle East and referred to as a remedy for many diseases and Habat Elbaraka: “the Blessed Seed” [1,2,3,4,5]. For many centuries in the Middle East, India, and North Africa, they were used for diverse ailments such as asthma, bronchitis, inflammatory conditions, hypertensive disease, diabetes mellitus, headache, fever, seizures, GI disturbances, and microbial infections. It is also used to stimulate appetite and increase breastfeeding mothers’ milk [6,7,8,9]. Nigella sativa contains 36 to 38% oil, proteins, alkaloids, saponins, and 0.4 to 0.45% volatile oil. TQ is the primary active substance found in the Nigella sativa seeds, which possesses several therapeutic benefits with minimal chances of toxicity or side effects [4,6,7,10]. TQ (Figure 1) is a monoterpene diketone with a molecular weight of 164.204 g/mol, and it is a lipophilic compound. With those characteristics of low molecular weight and being a lipophilic substance, it poses the ability to cross the blood-brain barrier. Thus, it is considered a potential substance targeting the central nervous system. Oxidation of quinone through heat will produce thymol, converted to Thymohydroquinone through controlled heating. Furthermore, the continuation of the heating process will result in the production of TQ [8,11].

Thymoquinone’s pharmacokinetic behaviors have been reported in different studies [12,13]. Animal models were used to investigate its pharmacokinetics following intravenous (5 mg/kg) and oral administration (20 mg/kg). Blood sample concentration of thymoquinone was measured using high-performance liquid chromatography, which was much higher following intravenous administration than oral administration (7.49 ± 0.24 vs. 3.48 ± 0.12 μg/mL), respectively [14]. Following intravenous administration, the clearance was 7.19 ± 0.83 mL/kg/min, and the volume of distribution was 700.90 ± 55.01 mL/kg [14]. While following oral administration, the clearance was 12.30 ± 0.30 mL/min/kg, and the volume of distribution was 5109.46 ± 196.08 mL/kg [14]. The calculated half-life (T_1/2_) following intravenous and oral administration were 63.43 ± 10.69 and 276.61 ± 8.48 min, respectively—the estimated thymoquinone bioavailability was ~58% [14].

TQ has several therapeutic effects on neurological disease, with the anticonvulsant, antianxiety development, antidepressant, antipsychotic action. Furthermore, it could be used in memory and cognitive disorders cases to treat drug abuse and dependence [8,15,16,17]. The variation in strokes among people regarding the degree of ischemia, localization, duration, and causes makes it one of the most challenging central nervous system (CNS) diseases [18]. The injury caused by ischemia-reperfusion in the brain is responsible for producing a considerable amount of ROS (hydroxyl radical, superoxide, and hydrogen peroxide) that leads to oxidative stress, and brain tissues are the most vulnerable to oxidative damage [19,20,21]. TQ, as an antioxidant, acts as a free radical scavenger or trapping agent and diminishes ROS-mediated reactions [4,7,19]. TQ has beneficial effects in preventing oxidative stress injury to brain cells, more prominent in the regions responsible for memory function [22]. TQ is also active on the CNS as antiepileptic. In AD, TQ mitigates stress-associated inflammation and is reported with a protective effect on neurons against Aβ-induced injury in PC12 cells [6,23,24]. TQ has protective effects against neurotoxin agents, which give the pharmacological properties of TQ and its implication in the protection against neurodegenerative diseases. It works to reverse the neurotoxicity produced by chemical agents such as lead, ethanol, toluene, glutamate, acrylamide, lipopolysaccharide, and streptozotocin [23,24]. TQ is also reported to protect from certain drugs such as morphine and pentylenetetrazol, which can produce toxicity, mainly in large doses. The previously mentioned agents are risk factors for causing CNS-related disorders that are considered neurodegenerative, such as AD, PD, dementia, epilepsy, etc. [24,25,26].

## 2. Effect of Thymoquinone on Neuroinflammation

The main etiologic factor responsible for the pathophysiology of AD and PD is neuroinflammation. It can be initiated or enhanced by responding to stimuli such as brain injury and infection. Enhancement of gene transcription can occur by nuclear factor Kappa-light-chain of activated B-cells, which act as a transcription factor that chains with DNA. Consequently, this enhancement of NF-kκB induces inflammation in CNS, especially in microglia, by releasing inflammatory mediators such as COX2 and iNOS and increasing the synthesis of Prostaglandins [27]. Furthermore, inflammation raises the cellular production of ROS. In this regard, TQ protects brain cells from different injuries and inflammation because it contains antioxidants and anti-inflammatory effects [28]. TQ treatment inhibits releasing TNF-a, mRNA, leukotrienes, including IL-6, IL-1b, and Prostaglandin’s synthesis. Essentially, TQ can inhibit the enzyme nitric oxide synthase (iNOS); as a result, it will reduce the nitrite (NO_2_^−^), which is evident by reducing the iNOS protein expression. Therefore, iNOS protein expression reduction can effectively alleviate inflammatory and autoimmune diseases.

Furthermore, in the case of activated microglia, TQ was found to reduce the pro-inflammatory cytokines and chemokines such as (IL-) 6, IL12p40/70, and granulocyte colony-stimulating factor [8,29]. TQ inhibits the enhancement of NF-KP activation and its binding with DNA, consequently inhibiting neuroinflammation. Moreover, the binding of nuclear factor erythroid-derived 2-like 2 (NFE2L2 or Nrf2) to the antioxidant responsive element (ARE) is enhanced by TQ treatment that inhibits the inflammation that NF-kκB causes microglia. LPS-induced inflammation of microglia is inhibited by TQ, which interferes with phosphoinositide 3-kinase (PI3K)/protein kinase B or NF-kκB signaling pathways [30,31]. Another neuroinflammatory pathway that TQ is reported to target is the AMP-activated protein kinase (AMPK) and nicotinamide adenine dinucleotide (oxidized)/sirtuin 1 (SIRT1) pathways [17]. TQ will activate the AMPK, which has several anti-inflammatory effects; one mechanism is increased expression of genes that have antioxidant effects. An additional mechanism is increasing the level of NAD+, which will activate SIRT1, which will deacetylate p65 and reduce NF-κB. Moreover, it will increase the accumulation of Nrf2, which is known as an antioxidant transcription factor. Another reported mechanism will reduce the LPS-induced ROS generation [17,32]. In experimental allergic encephalitis (EAE) rates, TQ eliminated the symptoms of the disease with no evidence of inflammation compared to rates not treated by TQ [17]. The primary mechanism that TQ mediates its anti-inflammatory properties is the significant decrease in NF-B and TNF production.

## 3. Effect of Thymoquinone on Neurological Diseases

### 3.1. Alzheimer’s Disease

AD is the most common neurodegenerative disorder manifested by reduced memory and other cognitive functions. The symptoms begin in the temporal and frontal lobes of the brain. AD usually affects patients above 60 years of age but sometimes those who are significantly younger. The primary mechanism for neural damage in AD is activating glial cells caused by the inflammation and releasing the proinflammatory cytokines in the hippocampus that can suppress memory and learning [33,34,35]. ACh is the principal neurotransmitter, and its declined release and subsequent hydroxylation with AChE and butyrylcholinesterase (BChE) decrease ACh rapidly, resulting in learning deficits [15]. Pathophysiology of AD is related to the poisoned consequence of beta-amyloid (Aβ), a dysfunctional amyloid precursor protein leading to the accumulation of amyloid plaques in the brain, resulting in damage to the signaling activity of the nerves [2,36,37]. Beta-amyloid (Aβ) peptides lead to neuronal damage through a complex cascade by releasing different types of proinflammatory molecules, such as ROS, TNF, and interleukin (IL-1b, IL-6), which cause mitochondrial damage and finally lead to apoptosis. Additionally, ROS can lead to macromolecules, e.g., lipids or proteins destruction and dysfunction in neuronal plasticity [33,38]. The complications of AD in the older population have resulted from the absence of cholinergic neurons in the basal forebrain and hippocampus regions. The duration of the symptoms is around 10 years [39]. TQ has a significant impact on retarding and preventing the progression of AD due to its anti-inflammatory and antioxidant properties. Reducing the effects of IL-6, TNF-α, NF-ķL, and the suppression of cytokine production can reverse the inflammatory effects of the hippocampus [33,40]. TQ can induce protection from Beta-amyloid (Aβ) peptides which cause neurotoxicity.

TQ ameliorates and prevents Aβ-induced neurotoxicity and mitochondrial membrane depolarization by inhibiting ROS formation and reducing oxidative stress by antioxidant properties. TQ improves synaptic vesicle recycling and inhibits Aβ 1–42 aggregation. Pretreatment with TQ can inhibit apoptosis and free radicals’ production, causing cell regeneration. In this regard, in vitro studies of anti-Alzheimer impacts of TQ demonstrate that TQ acting on signal pathways that TNF-a mediates leads to the inhibition of oxidation of Beta-amyloid (Ab). These happen through downregulation and upregulation effects on nitric oxide (NO) and glutathione (GSH) [41]. TQ has an antioxidant impact on BV-2 murine microglial cells, affecting ROS and proinflammatory cytokines whose expression is very high and significant in several neurodegenerative disorders. Moreover, the remedy with TQ of the lipopolysaccharide/interferon-gamma interacts with the expression of genes that participated in the oxidation process. In vivo studies show that pretreatment with TQ has significantly decreased the lack of hippocampal neurons. This effect is due to the antioxidant activity of TQ on the levels of the superoxide dismutase (SOD) and GSH activities. TQ has an essential role in the neuroprotective impact on hippocampal cells after cerebral ischemia through the inhibition of lipid peroxidation [42,43].

It has been demonstrated that in cases of overactivation of NMDA receptors and the release of the glutamine—which is also known to contribute to neuronal death and the progression of the disease by increasing the accumulation of Aβ 1–42 aggregates—TQ can reverse the effect of glutaminergic activation on neuronal cells by the inhibition of the apoptosis of neuronal cells in the hippocampus [44].

TLRs are receptors found in several immune cells, including astrocytes. It is a transmembrane protein with an extracellular leucine-rich repeat domain and an intracellular Toll–interleukin-1 receptor. The activation of TLR receptors will result in the activation of (NF-B) and subsequent release of other inflammatory mediators. TQ inhibits the TLR receptors signaling and, accordingly, inhibits the release of the inflammatory mediators [2,45]. Another proposed mechanism is the upregulation of the following four proteins: biliverdin reductase A (BVR-A), glutaredoxin-3 (Grx-3), 3-mercaptopyruvate sulfotransferase (3-MST), and mitochondrial Lon protease (LONM), which have been found to have a neuroprotective and antioxidant effect [46]. A summary of several animal-model studies on the effect of TQ on AD is illustrated in Table 1.

### 3.2. Parkinson’s Disease

PD is a multi-centric neurodegenerative disease distinguished by degeneration of the dopaminergic system in the pars compacta, the portion of the substantia nigra. The pathophysiology of PD is mainly related to oxidative stress and inflammation. The major manifestations of PD are bradykinesia, rigidity, numbness, limpness, and resting tremors, and are reduced with dopamine replacement therapies [51,52]. Excessive and continuous muscle contraction causes the rigidity that characterizes resistance to movement. Furthermore, the progression of the disease may lead specifically to dyskinesia. Other symptoms in the advanced stage of PD such as dementia, depression, autonomic failure, and sleep abnormality are also evident. Previous clinical therapy of patients includes observing several factors such as signs and symptoms, disease stage, age, and level of functional disability. Some medicinal herbs such as Nigella sativa improve PD symptoms and prevent the deterioration of motor symptoms [53,54,55]. Rotenone, an insecticide and pesticide, can cause movement failure or Parkinson’s symptoms such as incoordination, muscle tremor, or rigidity. Cotreatment of rotenone and TQ prevented PD symptoms induced by rotenone [56,57,58]. TQ prevents free radicals’ formation; as a result, cell damage due to oxidative agents is diminished.

The mitochondrial function must be preserved because it mediates apoptotic cell death in case of dysfunction. This preservation is reported with TQ. It also has a protective effect against MPTP which induces cell death of dopaminergic cells. MPTP and rotenone exert their neurotoxic influence through their oxidation injury, and the antioxidant activity of TQ can oppose the effects of those neurotoxic substances [46,59,60]. Table 2 shows the different animal studies that addressed the impact of TQ on PD.

A-SN oligomers are also related to the pathophysiology of PD. TQ protects rats’ hippocampus from a-SN-oligomers-induced synaptic toxicity [30]. TQ has the effect of restoring the synaptic vesicle recycling activity after exposure to A-SN [30]. The antipsychotic property of Haloperidol can cause extrapyramidal effects. These neuroleptic effects evoke the metabolism of dopaminergic neurons, leading to EPS through ROS production. Haloperidol also induces bilateral neuronal degeneration and astrogliosis. This degeneration is inhibited by NS containing TQ by decreasing ROS production and controlling the free radical formation [30]. Treatment with TQ results in a noticeable reduction in GFAP immunoreactive astrocytes. When GFAP is deceased with less astrocytic stimulation, it reduces striatal gliosis. Consequently, no symptoms indicated the loss of dopaminergic neurons in the treated group of animals with TQ. The effect can be attributed to the antioxidative property of TQ, which exhibits a neuroprotective effect through inhibition of astroglia-induced toxicity in neurons [30]. EPS of Haloperidol was not seen in rats who received TQ [65]. Catalepsy is induced by chlorpromazine, which increases oxidative stress and lipid peroxidation. After administration to the rats, there was an increase in catalepsy and a decrease in body weight and mobilization. Two significant factors were disturbed after Chlorpromazine administration. Firstly, it increases the level of thiobarbituric acid, responsible for lipid peroxidation, and increases nitrite. Secondly, it decrees the level of glutathione “GSH,” which naturally has antioxidant properties. Cotreatment with TQ, which has anticataleptic effects in chlorpromazine-treated rats, can decrease TBARS and nitrite levels. Additionally, the availability of GSH against oxidative stress is increased, demonstrating that TQ has antioxidant properties, showing a reduction in free radicals’ formation [64].

### 3.3. Epilepsy

Epilepsy is one of the most heterogeneous neurological disorders caused by a simultaneous electrical release of neurons in the brain and is considered a biochemical phenomenon that is not completely understood. It is characterized by persistent neuronal activity and repeated spontaneous seizures. Epilepsy is considered an asymptomatic occurrence instead of a disease arising from traumatic brain injury and genetic factors [66,67]. As a result of an increased escape of glutamate neurotransmitters, it leads to binding with glutamatergic neurons, giving rise to high liberation of calcium in the postsynaptic neuronal cells. Epilepsy has different categories that depend on age, type of seizures, deterioration of the condition, and therapy [68]. Cognitive and affective disorders have some drawbacks, such as impaired emotional learning and spatial memory deficit. Studies have shown that epilepsy destroys some brain structures, such as the hippocampus and limbic system [69,70]. TQ offers protection from glutamate-induced cellular toxicity in SH-SY5Y neurons. Glutamate has several toxic effects as it can cause loss of viability, the generation of ROS, dysfunction of the mitochondria that will lead to apoptosis through the decreased Bax/Bcl-2 ratio, and increased expression of caspase 9. TQ can also protect from the effects of glutamate by reducing ROS production by inhibiting mitochondria dysfunction, hence inhibiting apoptosis [69,71].

The impairment in the inhibitory-excitatory balance has been viewed as an essential mechanism in the pathogenesis of epilepsy. It is explained by increased glutamate function on the *N*-methyl-d-aspartate NMDA receptor and decreased GABA receptor activity. Activation of GABA receptors will lead to the influx of Cl ions, which will result in hyperpolarization and inhibition of neuronal activity. Picrotoxins and bicuculline are known inhibitors of the GABA receptors, and several studies have shown that TQ’s anticonvulsant activity involved Picrotoxins-sensitive and bicuculline-sensitive GABA receptors. In addition, TQ modulates its anxiolytic effects through nitric oxide-cyclic guanosine monophosphate (NO-cGMP) and the GABA pathway [30,72].

Pentylenetetrazol (PTZ) can induce apoptotic neuronal cell death of hippocampal rats leading to seizures with increased caspase 3 activity. PTZ also increases the number of apoptotic neuronal cell death in the cerebral cortex, which indicates that PTZ causes neuronal loss in that area. Apoptotic cell death is diminished by administering TQ mainly through the antioxidative property. The release of Cytochrome-C and the oxidative stress process are affected by the mitochondrial pathway, which impacts apoptosis. TQ might prevent apoptosis via altering mitochondrial function and lowering Cytochrome-C and Caspase-3. However, it increases the expression of Bcl-2, which preserves membrane potential and blocks cytochrome-C release. The number of neurons that contain GABA-A is elevated at the time of the TQ treatment [73,74]. Furthermore, it can ameliorate astrogliosis neurons. The antiepileptic effect of TQ is explained by decreasing seizure activity and oxidative degradation of lipids, hippocampal neuronal cell loss, and astrogliosis. There are significant effects of TQ on myoclonic seizure in PTZ treated rats. The effects are both the prolonged onset of a seizure and the decreasing of its duration [30]. TQ can increase GABAergic transmission via opioid kappa receptors, which usually affect Ca^2+^ channels and blocks cellular Ca^2+^ influx. This result shows that the anticonvulsant effect of TQ is produced via opioid kappa receptors. TQ prolongs the latency that shows the seizure onset at 44.4 s to 265.7 s and decreases the duration from 12.2 s to 5.8 s [66].

## 4. Effect of Thymoquinone on Learning and Memory

Learning and memory are essential for developing the human brain, and impairment is a significant cause of dementia. It can occur by aging, brain injuries, or neurodegenerative disorders. Usually, impairment interferes with learning and memory function. The effects of TQ on memory and cognition are related to spatial memory. This includes the ability of the brain to recognize, store, and recover information [75]. ACh, a neurotransmitter in CNS, plays an essential role in managing learning and memory. Memory diminishing could be caused when the release of ACh decreases. AChE is an enzyme used for the degradation of ACh. Clinical studies establish that TQ inhibits AChE activity, which increases ACh, thereby preserving the effects by programming new memories. Another study reports that TQ prevents memory deficit induced by scopolamine in rats, as the study showed a decrease in the release of AChE activity in the cortex tissue and hippocampus. TQ has a similar effect to donepezil, an inhibitor of AChE known to have a favorable impact on reducing brain tumor necrosis factor-alpha (TNF-α) content and increasing glutathione brain contents [76]. TQ has a valuable ability that protects the brain against cellular damage due to oxidative stress through free radical scavenging properties, which could help preserve memory loss. One study shows that prolonged use of TQ increases the consolidation and recall capability of information storing and spatial memory in diabetic animals [22].

## 5. Safety and Adverse Effects

The Nigella Sativa oil extracts appear to have a low toxicity level. The administration of 50 mg/kg of oil to the rats for five days did not show any meaningful hepatic and renal enzymes activity. When administrating doses up to 10 mL/kg in rats, there were no signs of toxicity during the 48 h of observation. This indication was also shown when oral administration for 12 weeks reported the same result as the 48 h observation. There was not any change in mortality or haptic enzyme. Acute administration of 2 g/kg or more, considered an extremely high dose, could cause respiration problems. Clinically, high doses release an antioxidant known as glutathione in the heart, kidney, and liver. Administration concentrations of up to 0.03% for 90 days of TQ in mice’s drinking water showed no toxicity. However, there was a significant decrease in the fasting plasma glucose concentration [77,78]. Furthermore, a concentration of 1 mM instantly caused cytotoxicity, as exhibited by nuclear shrinkage and plasma membrane blebs. TQ lower concentrations such as 100 lM cause cell death within a few hours after receiving the treatment. TQ concentrations of 50 lM and 25 lM displayed acute cytotoxicity by high necrosis levels and nearly complete cell annihilation at the time of collecting within two days. Lower concentrations inducing necrosis were less toxic. Increasing the concentration of TQ up to 10 lM induces genotoxicity, a significant increase in the reoccurrence of chromosomal aberrations. Furthermore, it shows cystic- and genotoxic effects in a concentration-dependent manner. At 2.5 lM, the cytotoxic effect is proven by an excessive increase in necrotic cells. At 20 lM, an antiproliferative effect is supported by a significant decline of mitotic cells [79].

## 6. Conclusions

TQ is the primary active crude extract of *Nigella sativa* seeds and shows many therapeutic benefits, lowers chance of toxicity, and presents minimized side effects. Researchers in different regions have studied the evolution of ancient uses of TQ to find an alternative remedy to the current treatments, and they work to minimize the side effects of the use of TQ. The previous studies showed many effects of TQ against neurological disorders, specifically the neurodegenerative diseases AD and PD. In addition, with its anti-inflammatory effects, TQ has neuroprotective effects in AD, epilepsy, and PD.

## Figures and Tables

**Figure 1 pharmaceuticals-15-00408-f001:**
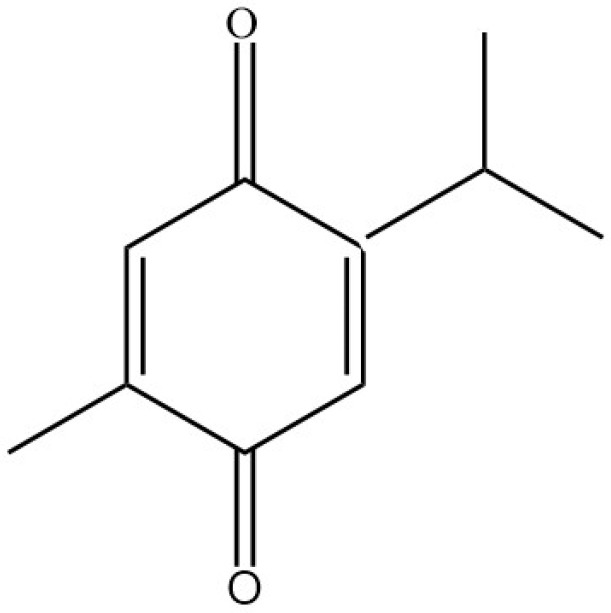
Thymoquinone chemical structure.

**Table 1 pharmaceuticals-15-00408-t001:** Studies on animal models for the effect of Thymoquinone on AD.

Disease	Animal Model	Treatment	Tissue Sample	Result	References
AD	48 male albino rats	Lipopolysaccharide with a dose of 0.8 mg/kg was given as an injection into the peritoneum for one dose. Group III was treated by a TQ 10 mg/kg injection into the peritoneum. Group IV was treated by PNU-120596 1 mg/kg injection into the peritoneum.	frontal lobe	More effective using TQ or α7 nAChR agonist and PAM.	[1]
AD	Male rats	D-gal dose of (60 mg/kg day) and AlCl3 dose of (10 mg/kg day) administered through the peritoneum (i.p.) once daily for 42 days, and after 4 weeks, TQ was administered intragastrically (i.g.) as a dose of (20 mg/kg/day) once daily for 14 days.	whole brain	Increased potential protective effect of TQ.	[2]
AD	Twelve-week-old male Wistar rats	Group (1) is the Control group received (saline). group (2) received LPS (1 mg/kg i.p.), groups (3–5) received 2, 5, or 10 mg/kg TQ treatment.	hippocampal and cortical tissues	Improved the impairment of learning and memory.	[33]
AD	Amyloid beta- (Aβ-) induced neurotoxicity	The intervention group received Aβ1–42 and TQ as a treatment simultaneously for 72 h.	hippocampal and cortical neurons	Efficient attenuation of Aβ1–42-induced neurotoxicity	[47]
AD	Adult female rats injected by STZ (3 mg/kg)	TQ dose of 20 mg/kg/day was given to rats for 15 days; on the 15th day, STZ injection was given.	hippocampus	Noticeable decrease in STZ-induced neurodegeneration.	[48]
AD	Thirty adult male Sprague Dawley albino rats	(Control group, Group 2 is people with AD): induced by oral AlCl3 (17 mg/kg/day) for 4 weeks. Group 3 (TQ/AD): treated with oral TQ (10 mg/kg/day) and AlCl3 (17 mg/kg/day) for period of 4 weeks.	hippocampus	Protective effects against neurodegeneration.	[49]
AD	Adult female rats injected with aggregated Aβ1–42	TQ dosage of (10 mg/kg) was given. The other group received a TQ dose of 20 mg/kg) for 15 days.	hippocampal tissue	Reduced neurotoxicity by removing Aβ plaques and restoring neuron viability.	[50]

**Table 2 pharmaceuticals-15-00408-t002:** Animal studies on the effect of Thymoquinone on PD.

Disease	Animal Model	Treatment	Tissue Sample	Result	References
PD	PD mouse model.	TQ (10 mg/kg was given for 1 week before administration of MPTP (25 mg/kg).	Striatal region	Inhibition effect against α-synuclein aggregation and cellular death.	[3]
PD	Primary dopaminergic cell culture neurons.	dopaminergic neurons tissue was received TQ (0.01, 0.1, 1, and 10 μM) on day 6 i.v. for 6 days.	NA	Protective effects against MPP+ and rotenone.	[56]
PD	Embryonic mouse mesencephala at gestation day 14.	Four groups: group 1 control group, group 2 received TQ on the 8th day for 4 days, group 3: received 1-methyl-4-phenylpyridinium (MPP+) on the 10th for 48 h, group 4:co-treated with TQ and MPP+.	NA	Protective effects on the dopaminergic neurons and inhibition of their apoptosis.	[61]
PD	6-hydroxydopamine (6-OHDA)-lesioned rats.	Oral TQ at different doses of 5 and/or 10 mg/kg administered 3 times daily for 1 week.	Substantia nigra pars compacta and midbrain	Protective effect against 6-OHDA neurotoxicity.	[62]
PD	Male Wistar rats (8–10 months) received rotenone.	TQ (7.5 and 15 mg/kg/day, po) given as pretreatment for one hour before administration of rotenone injection.	Substantia nigra (SN) and striatum (ST)	Protection and antioxidant effects against rotenone.	[63]
PD	Adult Wistar rats of either sex, CPZ dosing for 21 days to induce Parkinson’s.	Extracts of Nigella sativa at 200 and 400 mg/kg doses were given orally.	Whole-brain	Increased anti-Parkinson’s activity	[64]

## Data Availability

Data sharing is not applicable.

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
