# Peer review of "Thymoquinone: Review of Its Potential in the Treatment of Neurological Diseases"

_pharmaceuticals, 2022, doi:10.3390/ph15040408_

Round 1
Reviewer 1 Report
The work of Pottoo et al. Provides an overview of the potential of Thymoquinone in treatment of neuro- 2logical disorders.
1. Although the work contains many of the essential and succinctly presented properties of Thymoquinone, it lacks a few elements that will make the work more valuable.
2. You have to place the structure of the molecule and indicate which element of the structure will be responsible for its properties.
3. Present the pharmacokinetics of the compound in the organism and write down the concentration in which it works best for animals and humans.
4. At work, for example, drawings and graphs are needed, thanks to which the work would be more legible.
5. In tables 1 and 2, the results should be bulleted to make the table more readable.
6. The table should contain relevant information, not descriptions.
7. The work contains only the positive qualities of the relationship. What about negative properties?
Author Response
Respected Reviewer
Some comments were received in the paper entitled " Thymoquinone: Review of its potential in treating neurological disorders," some comments were received. I am incredibly thankful to the reviewer for showing keen interest and encouraging authors with positive feedback, thereby seeding motivation to improve this review article.
Kindly find the reply to comments in the tabular form below, while the text is incorporated into the main file (highlighted in yellow) at appropriate places.
|
S.NO |
Comments |
Author/s |
|
-Reviewer 1 |
||
|
01 |
You have to place the structure of the molecule and indicate which element of the structure will be responsible for its properties. |
The authors appreciate the comment of the reviewer, and the chemical structure of thymoquinone were added (Please refer to the introduction section (Fig.1) ).
|
|
02 |
Present the pharmacokinetics of the compound in the organism and write down the concentration in which it works best for animals and humans. |
The authors appreciate the important assertion made by reviewer, we have included the pharmacokinetics parameters of thymoquinone with the concentration (Please refer to the introduction section) |
|
03 |
In tables 1 and 2, the results should be bulleted to make the table more readable. |
Thanks for the valuable comment, the tables were adjusted. |
|
04 |
The table should contain relevant information, not description. |
Thanks for the valuable comment, the tables were adjusted. |
|
05 |
The work contains only the positive qualities of the relationship. What about negative properties? |
Safety and adverse effects of TQ have been mentioned in the section 5.
|

Reviewer 2 Report
this study answered the main question which aims to illustrate the importance of thymoquinone as a potential compound in treating the neurological disorders. This study is original and interesting. It summarizes and concentrates on the thymoquinone and its potential in treating the neurological disorders
minor: I have noticed that the name of the plant is not italic in most of the text such as in the abstract line 9 (Nigella sativa) and in the introduction line 1, 7, and 9. Please revise.
Author Response
COMMENTS FROM – Reviewer 2
Respected Reviewer,
We want to express our sincere gratitude and thanks to you for taking your precious time in reviewing the manuscript entitled " Thymoquinone: Review of its potential in the treatment of neurological disorders " In line with your positive feedback, we have addressed the comments.
Kindly find the reply to comments in the tabular form below, while the text is incorporated into the main file (highlighted in yellow) at appropriate places.
|
S.NO |
Comments |
Author/s |
|
|
-Reviewer 2 |
|||
|
01 |
This study answered the main question which aims to illustrate the importance of thymoquinone as a potential compound in treating the neurological disorders. This study is original and interesting. It summarizes and concentrates on the thymoquinone and its potential in treating the neurological disorders.
Minor: I have noticed that the name of the plant is not italic in most of the text such as in the abstract line 9 (Nigella sativa) and in the introduction line 1,7,and 9.please revise.
|
The authors appreciate the important assertion made by reviewer; we have made the changes. |
|

Reviewer 3 Report
This review is focused on pharmacological activity of Thymoquinone in neurological disorders as Alzheimer’s disease (AD), Parkinson’s disease (PD) and epilepsy.
The review is interesting, clear and well written, focused on the topics. The pharmacological properties of Thymoquinone are many and many references are reported in the literature (about 4,000 using SCIFINDER as data bank). Considering the IF of this journal, the authors should implement the paper, introducing more bibliographical references (in this version only 40 references are reported) and referring in more depth to the other pharmacological activities of the compound (antibacterial, anti-cancers and others).
However, the authors should also add the chemical structure of Thymoquinone.
Author Response
COMMENTS FROM – Reviewer 3
Respected Reviewer,
We would like to express our sincere gratitude and thanks to you for taking your precious time in reviewing the manuscript entitled " Thymoquinone: Review of its potential in the treatment of neurological disorders " In line with your positive feedback, we have addressed the comments.
Kindly find the reply to comments in the tabular form below, while the text is incorporated into the main file (highlighted in yellow) at appropriate places.
|
S.NO |
Comments |
Author/s |
|
|
-Reviewer 3 |
|||
|
01 |
The review is interesting, clear and well written, focused on the topics. The pharmacological properties of Thymoquinone are many and many references are reported in the literature (about 4,000 using SCIFINDER as data bank). Considering the IF of this journal, the authors should implement the paper, introducing more bibliographical references (in this version only 40 references are reported) and referring in more depth to the other pharmacological activities of the compound (antibacterial, anti-cancers and others). However, the authors should also add the chemical structure of Thymoquinone |
The authors appreciate the important comment made by reviewer; we have included 40 more updated bibliographical references. Regarding the other pharmacological references, we include the pharmacological activities which are related and focused on the neurological disorders. The chemical structure was added. |
|

Round 2
Reviewer 1 Report
Thank you for the changes made. Now the work is clearer .
Reviewer 3 Report
In this revised version paper can be published.